# Lycopene in Combination with Insulin Triggers Antioxidant Defenses and Increases the Expression of Components That Detoxify Advanced Glycation Products in Kidneys of Diabetic Rats

**DOI:** 10.3390/nu16111580

**Published:** 2024-05-23

**Authors:** Ingrid Delbone Figueiredo, Tayra Ferreira Oliveira Lima, Paulo Fernando Carlstrom, Renata Pires Assis, Iguatemy Lourenço Brunetti, Amanda Martins Baviera

**Affiliations:** 1Department of Clinical Analysis, School of Pharmaceutical Sciences, São Paulo State University (Unesp), Araraquara 14800-903, SP, Brazil; ingrid.delbone@unesp.br (I.D.F.); tayraferreira@yahoo.com.br (T.F.O.L.); fernando.carlstrom@unesp.br (P.F.C.); renata.assis@docente.unip.br (R.P.A.); iguatemy.brunetti@unesp.br (I.L.B.); 2Institute of Health Sciences, Paulista University (Unip), Araraquara 14804-300, SP, Brazil

**Keywords:** antidiabetic activity, glycoxidative stress, glyoxalase, paraoxonase, diabetic complications, natural bioactives

## Abstract

Background: Biochemical events provoked by oxidative stress and advanced glycation may be inhibited by combining natural bioactives with classic therapeutic agents, which arise as strategies to mitigate diabetic complications. The aim of this study was to investigate whether lycopene combined with a reduced insulin dose is able to control glycemia and to oppose glycoxidative stress in kidneys of diabetic rats. Methods: Streptozotocin-induced diabetic rats were treated with 45 mg/kg lycopene + 1 U/day insulin for 30 days. The study assessed glycemia, insulin sensitivity, lipid profile and paraoxonase 1 (PON-1) activity in plasma. Superoxide dismutase (SOD) and catalase (CAT) activities and the protein levels of advanced glycation end-product receptor 1 (AGE-R1) and glyoxalase-1 (GLO-1) in the kidneys were also investigated. Results: An effective glycemic control was achieved with lycopene plus insulin, which may be attributed to improvements in insulin sensitivity. The combined therapy decreased the dyslipidemia and increased the PON-1 activity. In the kidneys, lycopene plus insulin increased the activities of SOD and CAT and the levels of AGE-R1 and GLO-1, which may be contributing to the antialbuminuric effect. Conclusions: These findings demonstrate that lycopene may aggregate favorable effects to insulin against diabetic complications resulting from glycoxidative stress.

## 1. Introduction

Type 1 diabetes mellitus (T1DM) is an autoimmune disease that leads to the destruction of insulin-producing pancreatic beta cells, culminating in absolute insulin deficiency. The role of the insulin resistance in the pathophysiology of T1DM is not yet fully understood, but insulin resistance has been associated with an inefficient glycemic control, culminating in the need for a gradual increase in the insulin dose in patients with T1DM to achieve the glycemic control, which in turn has, as a main consequence, the risk of hypoglycemia [1,2].

Although insulin therapy is the best treatment option for T1DM patients, chronic insulinization may cause hypoglycemia episodes [3], increased body weight gain [4] and increased oxidative stress that results in endothelial dysfunction and inflammation [5]. Therefore, knowing the importance of insulinization in T1DM and the potential adverse effects that can be observed, the use of combined therapies between a reduced insulin dose with bioactive compounds having antidiabetic potential can be an interesting option to achieve glycemic control and to contrast diabetic complications [6,7]. Combined therapies between insulin and bioactive compounds of natural origin have recently been proposed as strategies to achieve an effective glycemic control and to mitigate long-term diabetic complications related to glycoxidative stress, in addition to prevent the adverse effects of long-term insulinization [8,9].

One of the main benefits of combining natural bioactives with classic therapeutic agents for DM is the potential to lessen the consequences of the metabolic memory. The introduction of natural bioactives able to inhibit the biochemical cascades triggered by the advanced glycation, oxidative stress and inflammation may be useful to minimize the diabetic complications [10,11]. Studies have demonstrated the antioxidant and antiglycation activities of combinations of natural bioactives with antidiabetic drugs, aiming to verify their effectiveness in mitigating DM complications [12,13].

Several studies have demonstrated the beneficial effects of carotenoids (β-carotene, lutein, zeaxanthin, astaxanthin, bixin, lycopene) on pathophysiological disorders and complications observed in obesity and DM [14,15,16]. Lycopene (Figure 1) is the main carotenoid found in red fruits, including tomatoes and other vegetables. It is a linear polyene hydrocarbon with 11 conjugated and two unconjugated double bonds [17]. The antioxidant properties of lycopene are mainly associated with its ability to capture reactive oxygen species (ROS) [18,19]. Lycopene also attenuates glycative stress by decreasing RAGE levels [20]. Previous studies from our laboratory have demonstrated that the association of lycopene with other natural bioactives, including curcumin [21] or with antidiabetic agents, including metformin [22] is capable of improving several disorders observed in DM, as well as to dampen oxidative stress, which makes lycopene an interesting option for ongoing studies in strategies of combined therapies to contrast diabetic complications.

This study aimed to investigate whether the combined therapy based on a reduced insulin dose with lycopene has favorable effects on the metabolic impairments observed in streptozotocin-induced diabetic rats. The ability of lycopene and insulin, alone or in combination, to stimulate the endogenous defenses involved in the detoxification of ROS and advanced glycation end products (AGEs) in kidneys of diabetic rats, decreasing albuminuria, was the focus of this study. 

## 2. Materials and Methods

### 2.1. Animal Model and Induction of DM

Male Wistar rats (Rattus norvegicus) with body weight values of 140–160 g (6 weeks old) were kept under light/dark cycle (12 h) and controlled conditions of temperature (23 ± 1 °C) and humidity (55 ± 5%) in the Bioterium of the Department of Clinical Analysis (FCF/UNESP). The animals had free access to water and diet (Presence Nutrição Animal, Paulínia, Brazil) throughout the 30 days of experiment. The animals remained under these conditions for 48 h before the beginning of the experiments. The experimental procedures were approved by the Ethics Committee in Animal Experimentation of FCF/UNESP, Araraquara (CEUA/FCF/CAr 13/2020).

The experimental model of T1DM was induced through an intravenous injection of streptozotocin (STZ; 40 mg/kg) prepared in 0.01 M citrate buffer (pH 4.5) and administered into the jugular vein of 12 h fasted rats. Isoflurane inhalation was used as an anesthetic method. Normal, non-diabetic rats were also fasted and received only citrate buffer (jugular vein). Rats with postprandial glycemia values about 350 mg/dL were used to constitute the diabetic groups. Diabetic rats were distributed into different experimental groups by matching the animals according to similar medium values of glycemia and body weight. The experiments were initiated three days after the STZ administration, by treating the animals with insulin (4 U/day or 1 U/day), lycopene (45 mg/kg/day) or with the association of 1 U insulin/day + lycopene.

### 2.2. Experimental Design

Five experimental groups of diabetic rats (*n* = 12/group) were composed as follows: (i) diabetic rats treated with yoghurt (DYOG); (ii) diabetic rats treated with 4 U/day insulin (DI4U) (standard treatment); (iii) diabetic rats treated with 1 U/day insulin (DI1U) (reduced insulin dose); (iv) diabetic rats treated with 45 mg/kg lycopene into yoghurt (DLYC); (v) diabetic rats treated with 1 U/day insulin + 45 mg/kg of lycopene (DI1U + LYC). Non-diabetic rats treated with yoghurt (NYOG; *n* = 12) were used as controls.

Due to its low solubility in water, lycopene-rich tomato extract powder (Lycopersicon esculentum Mill., 10.13% lycopene, Ningbo Vitax Biotechnology Company, Ltd., Ningbo, China) was incorporated into yoghurt (170 g yoghurt contains 6.8 g protein, 7.0 g total fat, 9.1 g carbohydrates, 126 kcal, Nestlé^®^, São Paulo, Brazil) using a homogenizer (27,000 rpm) during 90 s at ± 25 °C. Daily doses of 45 mg/kg lycopene were divided into two doses of 22.5 mg/kg lycopene administered twice a day (8:00 a.m. and 5:00 p.m.) for 30 days. Lycopene into yoghurt was administered orogastrically (gavage) in 0.5 mL yoghurt (totaling 1 mL/day), whose dose was chosen according to previous studies [13,21].

NYOG and DYOG rats received only yoghurt (vehicle). Insulin (4 U/day and 1 U/day) was administered subcutaneously as two doses of 2 U (DI4U) or 0.5 U (DI1U) at 8:00 a.m. and 5:00 p.m. Insulin (Humulin^®^ N, U-100, Eli Lilly do Brazil Ltd., São Paulo, Brazil) was prepared in sodium chloride (0.85%). NYOG, DYOG and DLYC groups received saline (0.85%) subcutaneously, while DI4U and DI1U groups received 0.5 mL of water orogastrically to expose all rats to the same manipulation conditions.

### 2.3. Physiological Parameters, Temporal Glycemia Levels and Experiment Conclusion

During the experimental period, all rats were kept into polypropylene boxes (*n* = 4/box). On 0 and 30 days of experiment, rats were kept into metabolic cages (individually) to evaluated food and water intakes as well as the urinary volume of each animal. The 24 h urine samples (collected from metabolic cages after 30 days of experiment) were stored at −80 °C for the semi-quantitative analysis of the urinary albumin levels. Body weight was monitored weekly throughout the experimental period. Body weight gain was obtained by subtracting body weight value at the end of the experiment (day 30) from the body weight at the beginning (day 0).

Temporal plasma glucose levels were monitored on days 0, 10, 20 and 30 of treatment in the plasma obtained from blood samples collected from the tail of each animal (2 h after the respective treatments). Fasting blood glucose levels were monitored after a period of 6 h of fasting, on the 28th day of treatment. On the 29th day, a specific blood collection was performed from the tip of the tail in Eppendorf microtubes containing EDTA disodium salt (40 µL of EDTA: 200 µL of blood) for the analysis of the glycated hemoglobin (Hb1Ac).

The experiment ended after 30 days of treatments. To investigate the changes in the Serine-473 phosphorylation levels of AKT (basal or insulin-stimulated) in the gastrocnemius muscles, rats were fasted for 5 h and then anesthetized (5 mg/kg xylazine/50 mg/kg ketamine). Insulin (5 IU/kg; stimulated group) or saline solution (basal group) was administered intraperitoneally. Gastrocnemius muscles were collected after 10 min of insulin or saline administration and immediately stored at −80 °C for further analysis. Rats were euthanized by decapitation (using guillotine) due to the volume of blood required for the biochemical analysis. Blood samples were collected to obtain plasma, which was used for the analysis of biochemical markers: glucose, triglycerides, total cholesterol, fructosamine, creatinine and uric acid; in plasma, the glycoxidative stress biomarkers (TBARS, fluorescent AGEs) and the activity of paraoxonase-1 (PON-1) were also evaluated. Retroperitoneal and epididymal white adipose tissues were removed and weighed. Kidneys were also removed and stored at −80 °C for further analysis of glycoxidative stress biomarkers (fluorescent AGEs, TBARS), antioxidant enzymes (SOD, CAT, GSH-Px) and the protein levels of components belonging to AGE detoxification machinery (GLO-1, AGE-R1).

### 2.4. Biochemical Parameters

Blood samples were collected one day before euthanasia (29th day of treatment) for the determination of Hb1Ac levels via high-performance liquid chromatography (HPLC) using the Premier Hb9210^TM^ equipment (Trinity Biotech, São Paulo, Brazil). Plasma levels of glucose, fructosamine, triglycerides, total cholesterol, creatinine and uric acid were determined using commercial kits (Labtest Diagnóstica S.A., Lagoa Santa, Brazil).

### 2.5. Biomarkers of Glycoxidative Stress and Antioxidant Defenses

Levels of TBARS (biomarkers of lipid peroxidation, LPO) were determined in plasma and kidney homogenates through reaction with thiobarbituric acid (TBA) [23]. Plasma supernatants and kidney homogenates were prepared as previously described [13,21]. TBA reacts mainly with dienes, including malondialdehyde, generating products (TBARS) whose levels are determined fluorimetrically (excitation and emission wavelengths of 510 and 553 nm, respectively). The standard used was 1,1′3,3′-tetramethoxypropane (Sigma-Aldrich, St. Louis, MO, USA). The results were expressed in μmol/L.

Fluorescent AGEs (biomarkers of advanced glycation) were determined in kidney and plasma samples that were prepared as described by Figueiredo et al. [13]. Plasma supernatants and kidney homogenates were used to estimate fluorescent AGEs measured fluorimetrically at excitation and emission wavelengths of 370 and 440 nm, respectively [24]. The results were expressed as arbitrary units (AU) of fluorescence/mg of protein. Plasma total protein levels were determined using the method of Lowry et al. [25] and used to correct the results.

Paraoxonase-1 (PON-1) was determined by monitoring the release of p-nitrophenol (405 nm), according to the method adapted by Assis et al. [21]. In plasma samples, p-nitrophenol is released after paraoxon hydrolysis by PON-1. The activity of PON-1 was expressed in units/liter (unit = μmoL paraoxon hydrolyzed/min).

### 2.6. Semi-Quantitative Analysis of Albuminuria Levels

The urinary albumin levels (semi-quantitative analysis) were determined according to Kakani et al. [26] with modifications by Inacio et al. [27] and adaptations for this study. The total protein concentration in 24 h urine samples (after 30 days of treatment) was measured by using the pyrogallol red method [28]. SpeedVac^®^ Concentrator equipment (Thermo Fisher Scientific, Waltham, MA, USA) (at room temperature; 12 h) was used to concentrate the 24 h urine samples (500 µL) to enable their application in polyacrylamide gels containing enough protein mass for the visualization of the albumin bands. After the concentration process, urine samples were resuspended in different volumes of Tris-HCl buffer (0.5 M; pH 6.8) to achieve the following condition: 1.33 µg protein/µL buffer; a new analysis of the total protein concentration was performed to confirm the established condition.

The urine concentrated samples were prepared in Laemmli buffer (pH 6.8) before electrophoresis. Samples having 7.5 µg of protein were subject to SDS-PAGE electrophoresis in 12% acrylamide/bisacrylamide gels. The electrophoresis was performed in two steps: (i) at a constant potential of 60 V (30 min) and (ii) 120 V (90 min). After electrophoretic separation, protein bands were visualized by staining gels with a methanol/glacial acetic acid solution having 0.01% Coomassie Blue, followed by destaining with methanol/glacial acetic acid solution. The gels were imaged and densitometric values were obtained using ImageJ 1.53a (National Institutes of Health, Stapleton, NY, USA). Pure bovine serum albumin (1.00; 0.50 and 0.25 µg/well; product number A6003, Sigma-Aldrich, St. Louis, MO, USA) was used as standard. The densitometric analysis of pure bovine serum albumin bands allowed for the elaboration of a standard curve. Results were expressed as μg albumin/7.5 μg total protein.

### 2.7. Antioxidants’ Defenses in Kidneys

Kidneys samples were prepared according to Figueiredo et al. [13], and supernatants were used to evaluate activities of SOD, CAT and GSH-Px. Total protein levels were determined using the Lowry method [21] to correct the results.

The activity of SOD was determined by monitoring the inhibition of nitroblue tetrazolium chloride (NBT) reduction [29]. Xanthine oxidase promotes the xanthine oxidation, generating the superoxide anion radical (O_2_^•−^), which reduces the NBT to a formazan. SOD catalyzes O_2_^•−^ dismutation, inhibiting the NBT reduction, which is monitored at 550 nm. Results were expressed as U/mg protein. 

The activity of CAT was monitored by hydrogen peroxide (H_2_O_2_) consumption [30], monitored at 230 nm. The results were expressed as mmol H_2_O_2_ consumed/min/mg protein.

The activity of glutathione peroxidase (GSH-Px) was determined by monitoring the oxidation NADPH to NADP+ (nicotinamide adenine dinucleotide phosphate) [31]. GSH-Px catalyzes the oxidation of GSH (glutathione) in the presence of H_2_O_2_. Oxidized glutathione (GSSG) is then reduced to GSH by GSH-Rd (glutathione reductase) with concomitant oxidation of NADPH into NADP+, monitored at 340 nm. The results were expressed as mmol oxidized NADPH/min/mg protein.

### 2.8. Western Blotting Analysis in Muscles and Kidneys

Gastrocnemius muscles and kidneys were homogenized in Tris-HCl buffer (50 mM, pH 7.4) containing NaCl, EDTA, detergents and protease and phosphatase inhibitors. Homogenates were centrifuged (11,900× *g*; 4 °C for 30 min) and the supernatants were used for the analysis of the protein levels [25]. Tris-HCl sample buffer (pH 6.8) containing glycerol, SDS, DTT and bromophenol blue was used in the supernatants.

Supernatant samples (30 µg of protein) were subjected to SDS-PAGE electrophoresis in 10–12% acrylamide/bisacrylamide gels. The electrophoresis was performed in two steps: (i) at a constant potential of 60 V (30 min) and (ii) 120 V (90 min) at room temperature. Proteins were electroblotted onto nitrocellulose membranes (0.35 mA; 4 °C for 3 h). Membranes were incubated overnight at 4 °C with primary antibodies (Appendix A). Binding of primary antibodies was detected by HRP peroxidase-conjugated secondary antibodies (Appendix A) and visualized with a chemiluminescent substrate prepared in Tris-HCl buffer (pH 8.6) containing luminol, 4-iodophenylboronic acid and H_2_O_2_. Chemiluminescent bands were captured (C-Digit Blot Scanner, LI-COR, Lincoln, NE, USA) and the intensities of the bands were measured using LI-COR Image Studio 4.0.

### 2.9. Statistical Analysis

Data are expressed as mean ± standard error of mean (SEM). Differences between experimental groups were analyzed by one-way analysis of variance (ANOVA) followed by the Student–Newman–Keuls test. Paired Student *t*-test was used for intragroup comparisons against day 0. Statistical differences were considered when *p* < 0.05. Analyses were determined using the GraphPad Prism 6.01 program (GraphPad Software, San Diego, CA, USA).

## 3. Results

### 3.1. Combined Therapy 1 U/Day Insulin + Lycopene Prevented Weight Loss and Polyphagia in Diabetic Rats

Rats that received STZ and were treated with vehicle (DYOG group) underwent a decrease in the body weight gain (Table 1) that were accompanied by losses in the weights of adipose and muscle tissues; conversely, kidney weights were increased (Table 2). In addition, DYOG rats had polyuria, polydipsia and polyphagia (Table 1). The treatment of diabetic rats with 4 U/day insulin (DI4U group) prevented the weight loss and all other disturbances related to T1DM (Table 1 and Table 2).

The treatment of diabetic rats with a reduced insulin dose, 1 U/day insulin (DI1U group) caused an increase in the body weight gain in comparison with non-treated diabetic rats (DYOG group); however, their weights were lower than those of NYOG and DI4U rats (Table 1). Although the weight of gastrocnemius muscle of DI1U rats either increased or remained at similar values than those of NYOG and DI4U groups, the weights of adipose tissues increased but remained lower than those of NYOG and DI4U rats (Table 2). Regarding food and water intakes and urinary volume (Table 1), 1 U/day insulin partially restored these parameters in diabetic rats. 

No significant improvements were observed in diabetic rats treated during 30 days with lycopene alone (DLYC group) regarding body and tissue weights and the physiological parameters. On the other hand, the combined therapy 1 U/day insulin + lycopene (DI1U + LYC) allowed diabetic rats to achieve body weight values similar to those of NYOG and DI4U rats, a better result than that of the isolated treatments (Table 1), which may have been due to the preservation of the weights of white adipose tissues and gastrocnemius muscles (Table 2). The inhibitory effects of 1 U/day insulin + lycopene on polyphagia were significant: after 30 days of treatment, DI1U + LYC rats achieved low values of food intake that were similar from those of NYOG rats (Table 1). On the other hand, the inhibitory effects of the combined therapy on polydipsia and polyuria were slight: after 30 days of treatment, water intake and urinary volume of DI1U + LYC rats were decreased in comparison to DYOG rats, but remained higher than values of NYOG and DI4U groups (Table 1).

### 3.2. Combined Therapy 1 U/Day Insulin + Lycopene Decreased Glycemia and Improved Insulin Sensitivity in Diabetic Rats

At day 0 (three days after the administration of STZ), diabetic rats began the experiment with postprandial glycemia levels of 380 mg/dL (Figure 1A). DYOG rats had a progressive increase in the postprandial glycemia that reached values of 565 mg/dL after 30 days that was reflected in the highest AUC values of postprandial glycemia (Figure 1B) and fasting glycemia (Figure 1C). Corroborating these findings, DYOG rats exhibited the most evident reduction in the insulin-stimulated phosphorylation levels of AKT in gastrocnemius muscles when compared to the corresponding values in NYOG rats (Figure 2), suggesting impairments in tissue insulin sensitivity. Diabetic rats treated with 4 U/day insulin had improvements in the postprandial glycemia levels throughout the experimental period (Figure 1A,B), as well as in fasting glycemia (Figure 1C) and in the AKT phosphorylation levels (Figure 2).

Although the treatment of diabetic rats with 1 U/day insulin led to improvements in the insulin-stimulated phosphorylation levels of AKT in muscles at the same magnitude as 4 U/day insulin, even reaching values similar to those of NYOG group (Figure 2), it is interesting to observe that the glycemia control throughout the experiment was not effectively achieved: the glycemia levels of the DI1U group were lower than the values of the DYOG group; however, they remained higher than the corresponding values of the DI4U and NYOG groups (Figure 1A).

Treatment with lycopene alone slightly reduced the glycemia of diabetic rats when compared to DYOG rats (Figure 1) and did not promote improvements in the AKT phosphorylation in muscles (Figure 2). On the other hand, the best results in terms of glycemia were found in rats treated with 1 U/day insulin + lycopene. The treatment of diabetic rats with 1 U/day insulin + lycopene caused an additive effect on glycemia reduction (postprandial and fasting glycemia), suggesting that lycopene + insulin was more effective than the isolated treatments (Figure 1), which is probably related to the significant increase promoted in the insulin-stimulated phosphorylation levels of AKT in the gastrocnemius muscles (Figure 2).

### 3.3. Combined Therapy 1 U/Day Insulin + Lycopene Decreased Dyslipidemia, HbA1c and Biomarkers of Glycoxidative Stress, and Increased Antioxidant Defenses in Plasma of Diabetic Rats

After 30 days of experiment, DYOG rats exhibited dyslipidemia, with high plasma levels of triglycerides and cholesterol (Figure 3A,B), increases in HbA1c (150%, Figure 4A), and higher plasma levels of fructosamine (176%, Figure 4B), fluorescent AGEs (82%, Figure 4C) and TBARS (76%, Figure 4D) than the NYOG rats. Additionally, the activity of the antioxidant enzyme PON-1 in DYOG rats was reduced by 25% (Figure 4E). The treatment with 4 U/day insulin avoided the dyslipidemia (Figure 3A,B) and the increases in biomarkers of glycoxidative stress, with the exception that HbA1c had a slight increase in DI4U rats when compared with the NYOG group (Figure 4A).

The treatment with 1 U/day insulin was not capable to improve the HbA1c (Figure 4A) and the fructosamine levels (Figure 4B) in diabetic rats, despite improving the plasma levels of cholesterol and triglycerides (Figure 3), TBARS, fluorescent AGEs and the activity of PON-1 (Figure 4C–E). Treatment with lycopene alone only improved the levels of fluorescent AGEs and the PON-1 activity (Figure 4C,E). On the other hand, the combined therapy 1 U/day insulin + lycopene proved to be more effective in comparison to the isolated therapies, as it was able to improve the lipid profile (Figure 3), to effectively decrease the levels of all markers of glycoxidative damage and increase the PON-1 activity (Figure 4).

### 3.4. Combined Therapy 1 U/Day Insulin + Lycopene Reduced Albuminuria in Diabetic Rats

In order to assess the renal function, plasma levels of creatinine and uric acid were measured. DYOG rats had significant increases in creatinine (25%) and uric acid (50%) plasma levels (Figure 5C,D). The semiquantitative analysis of the urinary albumin excretion performed by protein gel fractionation indicated that DYOG rats exhibited increased albuminuria (1.57 ± 0.13 μg albumin/7.5 μg protein) in relation to NYOG rats (0.65 ± 0.04 μg albumin/7.5 μg protein). Furthermore, the separation pattern of the urinary proteins in the electrophoresis gel relative to the DYOG group indicated that the protein band corresponding to albumin appeared as a band of larger size, slightly “lagging” when compared to the albumin band (Figure 5A,B). This pattern can be attributed, at least in part, to structural changes in albumin into the circulation resulting from glycoxidative stress, affecting its molecular mass due to post-translational modifications, including crosslinking with other peptides or small circulating proteins.

Treatments of diabetic rats with insulin (4 U/day or 1 U/day) were able to decrease the plasma levels of creatinine and uric acid (Figure 5C,D) as well as the urinary albumin excretion (Figure 5A,B) in relation to the corresponding values in DYOG rats.

Although the treatment with lycopene alone improved the plasma levels of creatinine and uric acid (Figure 5C,D), it did not improve the urinary albumin excretion of diabetic rats (Figure 5A,B). However, the combined therapy proved to be the most effective among all other therapies in promoting the decrease in albuminuria. Diabetic rats treated with 1 U/day insulin + lycopene had a 70% reduction in the urinary albumin excretion in comparison to the corresponding values in DYOG rats (Figure 5C,D). It is important to mention that the urinary albumin levels in the DI1U + LYC group were even lower than those observed in the control group (NYOG), showing the effectiveness of the combined therapy in maintaining the renal integrity of diabetic rats. The treatment with 1 U/day insulin + lycopene also decreased the plasma levels of creatinine and uric acid of diabetic rats (Figure 5C,D).

### 3.5. Combined Therapy 1 U/Day Insulin + Lycopene Stimulated Cytoprotective Mechanisms Related to AGE and ROS Detoxification in Kidneys of Diabetic Rats

In addition to the impairments in plasma antioxidant enzyme PON-1, detrimental impacts were also observed in other components involved in the detoxification of ROS and AGEs in kidneys of diabetic rats. The activities of the antioxidant enzymes SOD (Figure 6A) and CAT (Figure 6B) were decreased, which probably explain the increases in the levels of TBARS, biomarkers of lipid peroxidation damage. The components of AGE detoxification system, including AGE-R1 (Figure 7A,B) and GLO-1 (Figure 7A,C) were also significantly lower in kidneys of DYOG rats than the corresponding values of NYOG rats. The treatment of diabetic rats with 4 U/day insulin (DI4U group) prevented all these alterations in the components of ROS and AGE detoxification and TBARS levels. It is interesting to note that in the kidneys of diabetic rats submitted to all treatments, there were increases in the activities of GSH-Px similar to the values observed in the DYOG group (Figure 7C). In addition, the fluorescent AGE levels were decreased and similar to all groups composed of diabetic rats (Figure 8B).

Although the treatment of diabetic rats with 1 U/day insulin partially restored the renal TBARS levels (Figure 8A), this improvement was not accompanied by recovering in the activities of the antioxidant enzymes SOD or CAT (Figure 6A,B). Furthermore, DI1U rats also did not have improvements in AGE-R1 levels (Figure 7A,B), despite having shown improvements in GLO-1 at levels similar to those from DI4U rats (Figure 7A,C). Treatment of diabetic rats with lycopene alone did not cause any improvement in AGE and ROS detoxification components, with the exception of the increases in the GLO-1 levels (Figure 7A,C).

Once again, a prominent response was achieved when 50 mg/kg lycopene were administered in combination with 1 U/day insulin to diabetic rats. The activities of SOD and CAT were significantly increased in kidneys of diabetic rats treated with 1 U/day insulin + lycopene (Figure 6A,B), which was not observed with the isolated therapies. The increases in the activities of the antioxidant enzymes in kidneys of DI1U + LYC rats were accompanied by decreases in the TBARS levels (Figure 8A). Furthermore, the highest increases in the levels of AGE-R1 (Figure 7A,B) and GLO-1 (Figure 7A,C) were achieved through the treatment with 1 U/day insulin + lycopene. Collectively, these findings indicate that this combined therapy triggers endogenous machinery that detoxifies ROS and AGE, which may be useful to dampen the glycoxidative stress in kidneys of diabetic rats.

## 4. Discussion

Insulin replacement is necessary for patients living with T1DM to achieve an effective glycemic control and to avoid the long-term complications resulting from glycoxidative stress; nonetheless, insulinization can bring some adverse effects, especially hypoglycemic episodes [32]. Therefore, the present study provides insights regarding the mechanisms underlying the protective effects of a therapeutic strategy based on reducing the dose of insulin and its association with a natural bioactive having antioxidant and antiglycation activities: lycopene. The main effects achieved with the combined therapy having lycopene and insulin at a reduced dose in diabetic rats were the following: (i) an effective glycemic control was achieved with the combined therapy, which was better than the isolated treatments and reached levels that were similar than those of normal, non-diabetic rats, which is probably due to the recovery of the insulin sensitivity; (ii) the combined therapy decreased the dyslipidemia and increased the PON-1 activity; (iii) the combined therapy exerted an antialbuminuric effect, which may be a consequence of the increased expression of the components related to AGE and ROS detoxification in the kidneys of diabetic rats.

In addition to the risk of hypoglycemic episodes, the intensive insulin treatment to insulinopenic individuals has been associated with the development of insulin resistance [1]. Furthermore, Akoumianakis and colleagues [5] observed that insulin treatment induces oxidative stress and endothelial dysfunction in the vessels of diabetic patients with atherosclerosis, which appears to be associated with insulin resistance. On the other hand, in this same study, the authors observed that the treatment with dipeptidyl peptidase 4 (DPP-4) inhibitors was able to restore the insulin sensitivity and to improve the redox state and the endothelial function, proving the importance of the effect of the combined treatment with insulin on cardiovascular outcomes in diabetic patients in terms of the clinical benefits.

Combined therapies between insulin and natural bioactives having antidiabetic, antioxidant and antiglycation activities have been proposed as strategies to achieve effective glycemic control, as well as to mitigate long-term diabetic complications related to glycoxidative stress [8,9,12,33,34]. Our laboratory has gained evidence about a wide range of benefits that were achieved when lycopene was combined with metformin to contrast the complications of both diabetes [13] and obesity [22]. In this present study, we advanced in understanding the role of lycopene in a combination strategy with a reduced dose of insulin.

Before we describe the effects of lycopene + 1 U/day insulin in the context of the glycemic control, it is important to highlight the beneficial effects of the combined therapy on the preservation of adipose and muscle tissue weights, which certainly contribute to explain its ability to prevent the weight loss in diabetic rats. There are reports of adults with T1DM that omit or restrict the insulin dose to lose or to control the body weight, which may lead to a poor metabolic control and can favor the development of short- and long-term complications resulting from hyperglycemia [35]. When observing the effects of the isolated therapies, it can be noted that the treatment with lycopene alone was not able to improve the weights of adipose and muscle tissues in diabetic rats. The treatment with 1 U/day insulin improved the tissue weights, but its effects were still smaller than those of the standard treatment, 4 U/day insulin. On the other hand, the treatment with lycopene + 1 U/day insulin had a synergistic effect, increasing the weights of adipose and muscle tissues, having a better response than those of the isolated treatments and achieving the effect of the standard treatment, 4 U/day insulin. Considering that the insulin sensitivity was significantly improved in gastrocnemius muscles of DI1U + LYC rats, this effect probably favored the anabolic responses of insulin.

Conversely, it is interesting to mention that two symptoms commonly observed in diabetic individuals were not improved with the use of the combined therapy: polyuria and polydipsia, although polyphagia was significantly attenuated. Although glycemia improved significantly in diabetic rats treated with lycopene + 1 U/day insulin, it is possible that small glycemic variations throughout the day culminate in episodes of glycosuria, which could explain the polyuria observed in DI1U + LYC rats, and consequently polydipsia. Another hypothesis is that other solutes are being eliminated in the urine of DI1U + LYC rats in significantly high amounts, including ketone bodies and urea, for example.

One of the most important findings of this study is related to the effective achievement of the long-term glycemic control in animals having T1DM and treated with lycopene + 1 U/day insulin; this glycemic control was better than those observed with the isolated therapies, and similar to the glycemic control of the group receiving the standard treatment, the DI4U group, and also similar to the control group, the non-diabetic, NYOG group. In addition, the DI1U + LYC group did not develop hypoglycemia. This effective long-term glycemic control could be certainly associated with the maintenance of tissue insulin sensitivity, an effect that may have arisen because of the beneficial effects of lycopene on preserving insulin signaling, as previously observed [36,37]. In improving the glycemic control, decreases in plasma glycoxidative stress markers were observed: the HbA1c levels were decreased, along with the early (fructosamine) and the advanced (AGEs) glycation biomarkers and the plasma lipid peroxidation markers (TBARS). Some of these markers, such as fructosamine, TBARS and HbA1c, had not been significantly improved with the isolated therapies, but were significantly improved in diabetic animals treated with lycopene + 1 U/day insulin, suggesting that the combined therapy had synergistic effects on markers of glycoxidative stress in plasma.

The beneficial effects of lycopene + 1 U/day insulin on promoting a decrease in the cholesterol and triglyceride plasma levels and increases in the PON-1 activity in diabetic rats also deserve to be highlighted. By decreasing the triglycerides and the cholesterol levels and increasing the PON-1 activity, the combined therapy may have atheroprotective effects on diabetic rats, a well-known effect attributed to carotenoids [38,39]. Furthermore, the antiatherogenic role of lycopene, which has been attributed to its antioxidant capacity that prevents the LDL oxidation in addition to its antihyperlipidemic effects, shed light onto its potential to be used in combination therapies to prevent cardiovascular diseases, for example, its combination with statins [40].

Although the lycopene effects on reducing the circulating lipid levels [41,42] are well known, including in a previous study from our laboratory [21], as well as its effects on reducing the levels of markers related to lipoperoxidation [43] and advanced glycation [44], in the present study, such effects were not observed. A plausible explanation for this lack of beneficial effects of lycopene alone may be related, at least in part, to the severity of the experimental model of T1DM, which certainly requires higher doses of the carotenoid.

An important advance achieved in the present study was the understanding of the renoprotective effects of lycopene + 1 U/day insulin in diabetes. The activities of the antioxidant enzymes SOD and CAT were completely restored in the kidneys of diabetic rats treated with the combined therapy. It is interesting to note that, with the isolated treatments, DI1U and DLYC groups, as well as in the untreated diabetic group, DYOG, there were no improvements in the activities of these antioxidant enzymes. Especially for the treatment with lycopene alone, it is important to observe that it did not promote significant improvements in the glycemia and in glycoxidative stress parameters in plasma. In addition, the treatment with 1 U/day insulin was less effective in the glycemic control in comparison to 4 U/day insulin. It is reasonable to suppose that the less effective antihyperglycemic effects of lycopene or 1 U/day insulin, alone, have contributed to the absence of their protective effects on the antioxidant enzymes; it is known that glycation processes inactivate antioxidant enzymes [45]. On the other hand, with the combined therapy, achieving the effective glycemic control certainly contributed to the preservation of the activities of the antioxidant enzymes in the kidneys of diabetic animals. Consequently, low levels of TBARS in kidneys were observed. Furthermore, it is known that lycopene increases the gene expression of antioxidant enzymes via the activation of the transcription factor Nrf-2 (nuclear factor (erythroid-derived 2)-like-2 factor) [46,47]. Albrahim and Robert [48] observed that rats fed a high-fat diet and treated with 25 or 50 mg/kg lycopene for 3 months resulted in preservation in the antioxidant enzymes SOD, CAT and GSH-Px in kidneys due to the stimulation of Nrf-2, decreasing renal injury.

Diabetic rats treated with lycopene + 1 U/day insulin also had significant increases in the expressions of GLO-1 and AGE-R1 in kidneys. GLO-1 participates in the AGE detoxification system that eliminates methylglyoxal [49]. The accumulation of methylglyoxal and methylglyoxal-derived AGEs in the kidneys has been associated with several damages, including collagen modification, mesangial expansion, glomerular basement thickening and adverse glomerular remodeling, leading to a decline in the glomerular filtration rate and increase in the albuminuria [50]. AGE-R1 receptor is postulated to mediate the uptake and degradation of AGEs by the kidneys, which facilitate their clearance into the urine [51]. Both GLO-1 [52] and AGE-R1 [51] are postulated to be down-regulated by diabetes, as observed in this study. Specifically regarding the expression of AGE-R1 in the kidneys, a study by He and collaborators [53] observed that the underexpression of this receptor occurs in mesangial cells of nonobese diabetic (NOD) mice; NOD mice are known to have features from both type 1 and type 2 diabetes [54]. This leads us to suggest that AGE-R1 expression is regulated by insulin, which could explain, at least in part, the significant increase observed in AGE-R1 expression in diabetic rats treated with lycopene + 1 U/day insulin. Furthermore, recent evidence indicates that AGE-R1 expression is also regulated by Nrf-2 [55]. Considering that some biological actions of lycopene occur via the activation of Nrf-2 [46,47], this reinforces the hypothesis that part of the effects of lycopene + 1 U/day insulin therapy on increasing the expression of AGE-R1 may be attributed to the activation of this transcription factor.

By increasing the expression of components that detoxifies AGEs and ROS, lycopene + 1 U/day insulin significantly decreased the urinary albumin excretion in diabetic rats. In fact, the urinary albumin excretion in DI1U + LYC rats was significantly decreased, reaching values lower than those observed in diabetic rats subjected to the standard treatment (DI4U group) and even lower than values of the normal, non-diabetic group (NYOG group). To the best of our knowledge, this study shows the first evidence of lycopene combining with insulin increasing the expression of GLO-1 and AGE-R1 in the kidneys of diabetic rats, decreasing albuminuria, which could be relevant for dampening glycoxidative stress.

The searching for new potential agents that induce the expression of components belonging to the cytoprotective mechanisms related to AGE and ROS detoxification, mainly the GLO-1 inducers [56], has been encouraged; interestingly, it has been demonstrated that Nrf-2 activators are GLO-1 inducers [57]. This hypothesis reinforces the need for further studies attempting to investigate the involvement of Nrf-2 activation in the mechanism of action of lycopene to stimulate the expression of cytoprotective components related to AGE and ROS detoxification, mainly SOD, CAT, GLO-1 and AGE-R1 in kidneys of diabetic rats.

## 5. Conclusions

This study demonstrated that lycopene, administered in combination with a reduced insulin dose, effectively controlled the glycemia of type 1 diabetic rats without causing hypoglycemia, decreasing glycoxidative stress, and in parallel decreased dyslipidemia and increased the PON-1 activity, which may be interesting to improve cardiovascular diseases. Furthermore, this combined therapy also triggered cytoprotective mechanisms that counteract the harmful impacts of glycoxidative stress in kidneys of diabetic rats, by increasing the expression of antioxidant enzymes (mainly SOD and CAT) and the components that detoxify the products of advanced glycation and/or their precursors (mainly GLO-1 and AGE-R1), which may be contributing to decrease albuminuria. Our results may contribute to a better understanding of the lycopene benefits in contrast to the renal complications of diabetes mellitus resulting from glycoxidative stress, especially in strategies of combined therapies with antidiabetic drugs.

## Data Availability

The datasets generated during and/or analyzed during the current study are available from the corresponding author on reasonable request due to privacy.

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
