# Peer review of "Lycopene in Combination with Insulin Triggers Antioxidant Defenses and Increases the Expression of Components That Detoxify Advanced Glycation Products in Kidneys of Diabetic Rats"

_nutrients, 2024, doi:10.3390/nu16111580_

Round 1

Reviewer 1 Report

Comments and Suggestions for Authors

General comments

The manuscript entitled “Lycopene in combination with insulin triggers antioxidant defenses and increases the expression of components that detoxify advanced glycation products in kidneys of diabetic rats” by Ingrid Delbone Figueiredo et al examined the lycopene administration on the diabetic rats. In this study, the authors carried out the trial to examine the synergic effect of dose (insulin) and lycopene.

The authors carried out many experiments using the tissues of the animals in the experiments by measuring some biological parameters. The purpose of the trial is interesting, as authors pointed out in the manuscript, to reduce the dose (insulin) is a quite important point to keep the high-quality life (not fully controlled by insulin).

However, several points are still unclear in the manuscript.

1, The authors used lycopene in the study. Lycopene is a long-chain hydrocarbon bearing some conjugated and unconjugated double bonds in the molecule. The structure is completely different from that of insulin. The reason for the choice of lycopene in the study is not clear.

2, Concerning the above point, other carotenoids or xanthine derivatives are also interesting to discuss.

3, It is very important to show the chemical structure of Lycopene for the better understanding of readers.

4, In the results section the authors showed the experimental results with brief explanation. The experiments were done six types, however, the explanation (sentences) is quite poor compared with the amount of experimental results

5, In the discussion part, the authors did not discuss their experimental results precisely. Namely, the authors discuss the experimental results by quoting several literatures. It is quite important point, however, authors should discuss their original experiments.

Tables

In Table 1, some physical parameters are measured, (body weight, body weight gain, food intake, water intake and urinary volume). It is very clear to find big differences between the negative control (NYOG) and positive control. It is also clear, the administration of 4U insulin (DI4U) improves the values close to NYOG. In the case of DI1U+LYC, the body weight gain is almost same with that of DI4U, however, water intake and urinary volume of DI1U+LYC showed significant differences with those of DI4U. The authors did not discuss these points (including other DYOG, DI1U, and DLYC studies).

In Table 2, the authors examined the weights of the tissues of rats. As found in the Table 1, DI4U showed the similar values (gastrocnemius muscle, Epididymal adipose tissue, Retroperitoneal adipose tissue, and kidneys). DYOG showed significant difference values with those of NYOG. In the case of DLYC, except kidney values, the obtained values are similar to those of DYOG. The values of DI1U and DI1U+LYC showed the values between DI4U and DYOG.  Comparing the DI1U and DI1U+LYC values, DI1U+LYC showed the closer value to DI4U than those of DI1U. These experimental results are quite interesting to discuss to investigate the biomarker studies.

Figures

In Fig. 2, the authors carried out the basal and insulin-stimulated phosphorylation levels of Akt in gastrocnemius muscles of diabetic rats. The sensitivity of DYOG is lower than NYOG, DI4U, DI1U, and DI1U+LYC. The authors got the results based on the densitometric analyses not q-PCR method. In the cases of insulin (-), DI1U showed higher value than that of DI4U. Is it really true? In this meaning, it is highly recommended to use the q-PCR for the study, as far as the authors discuss the qualitative work, it is meaningful to use densitometry, however, to discuss precisely, it is not adequate to use densitometric values.

In Fig. 3, the amount of triglyceride and cholesterol was increased in the case of DLYC. In Fig. 4, the amount of glycated hemoglobin, fructosamine, and TBARS in blood and plasma is increased in the case of LYC. These results suggest that lycopene does not act as a good antioxidant. While, by the combination with 1U insulin, the corresponding values became lower (almost same as 1U insulin). The explanation for these events must be required.

In Fig. 5 (B), the authors drew the urinary albumin amount in Y axis, and indicated significant different marks. The administration of DLYC increase the albumin amount compared with NYOG, D14U, and D1U. The combination of DI1U + DLYC decreased the value lower than those of DI4U, DI1U and NYOG. It is quite interesting observation, however, no explanation for this difference is not shown in the manuscript. The significant difference of DI1U+DLYC have three marks, b, d, e, is it really true? The DI4U value is bigger than DI1U value, so, b, c, d, e (or a, b, c, d, e) might be reasonable.

In Fig. 6, the authors carried out antioxidant enzymes insulin treated or non-treated rats. In the cases of SOD and CAT, DYOG, DI1U, and DLYC showed the lower values than those of NYOG, while DI4U and DI1U +LYC showed similar values with those of NYOG. The explanation for these events are not shown in the manuscript. In Fig. 5c, the authors indicated the significant different marks of DYOG, DI4U, DI1U, DLYC, and DI1U +LYC as b, is it really true. According to the footnote, mark b showed the significant difference versus DYOG.

In Fig. 7, the significant increase of AGER-1/ beta-actin was observed in the case of DI1U + LYC (not DLYC in the figure), the clear explanation for this might be necessary.

In Fig. 8 (b), the fluorescent AGEs (kidneys) were measured and similar values were observed in the cases of DYOG, DI4U, DI1U, DLYC, and DI1U+LYC. However, significant differences were found with that of NYOG. The authors carried out the similar experiments using blood sample (Figure 4C). In this case, only DYOG showed the significant difference (high score) to other samples. These tow results showed the different trend, it might be necessary to explain this event.

In the discussion part, the authors quoted many other researcher’s works to strengthen their logic, it is also quite important point. However, at first, the authors should scrutinize the experimental result, and cross-talk the experimental results precisely in the discussion section. Based on these, important literatures must be quoted to deepen the author’s logic. The combination of the bioactive compound(s) with bioactive substrate (hormone) is quite important topic to reduce the dose. There are so many choices, so clear logic for the use of bioactive compound(s) should be clarified.

Comments on the Quality of English Language

The English usage must be improved. It is very careful to prepare the manuscript before submission. Some grammatical mistakes (not so much) are found in the manuscript. In this meaning, the advice and revision of highly specialized person must be necessary.

Reviewer 2 Report

Comments and Suggestions for Authors

The study presented by Figueiredo and co-workers, explores the potential synergistic effects of combining lycopene, with reduced insulin doses to mitigate diabetic complications. By investigating its impact on glycemia and glycoxidative stress in the kidneys of diabetic rats, the research sheds light on promising therapeutic strategies. So, for this reason, I believe the paper being suggested for review is actual and really interesting. It's likely to grab the attention of many scientists.

The findings reveal that the combination of lycopene and insulin effectively controls glycemia, likely by improving insulin sensitivity. Moreover, this combined therapy ameliorates dyslipidemia and enhances the activity of paraoxonase 1 (PON-1), a significant marker of oxidative stress. In the kidneys, the treatment boosts the activity of antioxidant enzymes and modulates the levels of key proteins associated with glycoxidative stress, suggesting potential renal protective effects.

The introduction is short and at the same time comprehensive enough to introduce the reader to the topic of the problem.

In the square brackets, please leave a space between the comma and the next reference number – In all manuscript!

 The Materials and methods part is written in a proper and professional manner with details.

Row 509 a dot is missing at the end of the sentence.

Having thoroughly familiarized myself with the proposed article, I can confidently say that the conducted research is at a very high level, which can be confirmed by the presented methods and procedures, as well as by the obtained experimental data. I have no objections to the scientific value and discussion of the authors. The comments are completely clear and precise. Apart from the minor technical points I mentioned above, I have no critical comments and no recommendation.

For this reason, I recommend the authors correct the minor technical points.

Best regards,
